# Connecting metal-organic framework synthesis to applications using multimodal machine learning

Sartaaj Takrim Khan ◉ & Seyed Mohamad Moosavi ◉ ✉

Every year, researchers create hundreds of thousands of new materials, each with unique structures and properties. For example, over 5000 new metal-organic frameworks (MOFs) were reported in the past year alone. While these materials are often synthesized for specific applications, they may have potential uses in entirely different domains. However, linking these new materials to their best applications remains a significant challenge. In this study, we demonstrate a multimodal approach that uses the information available as soon as a MOF is synthesized, specifically its powder X-ray diffraction pattern (PXRD) and the chemicals used in its synthesis, to predict its potential properties and uses. By self-supervised pretraining of this model on crystal structures accessible from MOF databases, our model achieves accurate predictions for various properties, across pore structure, chemistry-reliant, and quantum-chemical properties, even when small data is available. We further assess the robustness of this method in the presence of experimental measurement imperfections. Utilizing this approach, we create a synthesis-to-application map for MOFs, offering insights into optimal material classes for diverse applications. Finally, by augmenting this model with a recommendation system, we identify promising MOFs for applications that are different from the originally reported applications. We provide this tool as an open source code and a web app to accelerate the matching of new materials with their potential industrial applications.

By selecting metals and organic linkers, we can potentially synthesize millions of possible metal-organic frameworks (MOFs)[1]. Given this high chemical tunability, MOFs can be tailor-made for various applications. MOFs have been explored extensively over the past two decades for applications ranging from catalysis[2] and gas separation[3] to sensing and electronics[4], and over 15,000 porous[5,6] and 120,000[7] nonporous MOFs have been synthesized. With the integration of automation and artificial intelligence (AI) in chemical synthesis[8–10], we can expect the discovery rate of new MOFs to grow even further.

Despite this significant progress, connecting newly synthesized materials to their best possible applications remains a challenge[11]. Determining the properties of a material requires extensive characterization and testing, often demanding expertise, resources, and infrastructure that may not be available to the researchers that synthesized the material. This hinders the realization of the full potential of new materials. There are several examples of materials that were found to be most effective for applications other than initially intended applications. Al-PMOF, initially synthesized for its photocatalytic properties[12], was only found years later to be highly effective at separating $CO_2$ from wet flue gases[13]. Similarly, SBMOF-1, originally created for $CO_2$ capture[14], turned out to be exceptional at separating Xenon from Krypton. These remarkable rediscoveries were enabled by high-throughput computational screening and machine learning studies years after the initial study were published[11,13,15–17].

Chemical Engineering & Applied Chemistry, University of Toronto, Toronto, ON, Canada. ✉e-mail: mohamad.moosavi@utoronto.ca

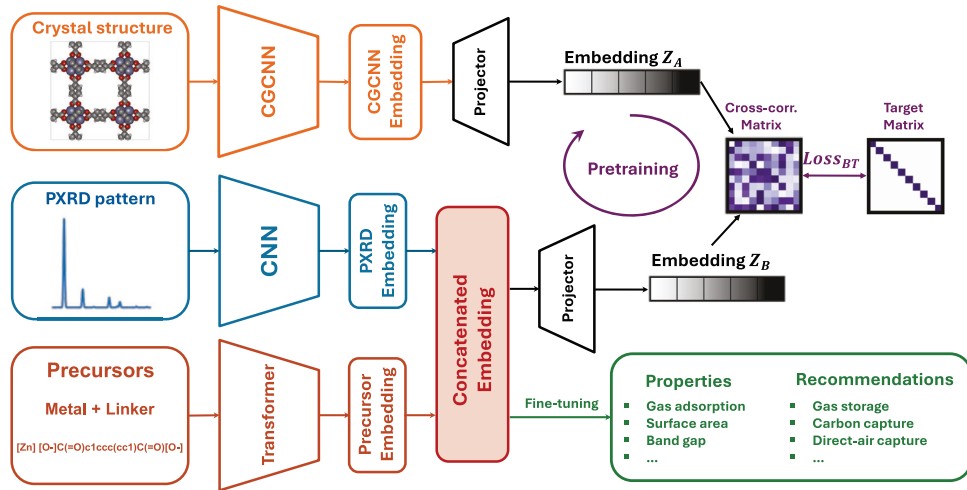

**Fig. 1 | Workflow of the self-supervised multimodal model.** The model takes a precursor string and powder x-ray diffraction (PXRD) spectrum as inputs, embedding them via a transformer and convolutional neural network (CNN), respectively. These embeddings are concatenated and passed through a regression head for fine-tuning. Prior to this, the model is pretrained using unlabeled crystal structures via a crystal graph convolutional neural network (CGCNN)[25]. The CGCNN and our model generate embeddings ($Z_A$, $Z_B$), which are used to construct a cross-correlation matrix. The Barlow-Twin loss ($Loss_{BT}$) minimizes the differences between the cross-correlation matrix and the identity matrix[23,52,53]. to align representations. The pretrained model is then fine-tuned on labeled data for property prediction and metal-organic framework (MOF) application recommendation. The crystal structure is visualized using iRASPA[55].

However, such methods require precise crystal structure information, often in a computation-ready format, which is complex to obtain and generally not available immediately after a new MOF is synthesized[5,18,19]. Therefore, developing methods that use only the data available upon synthesis can greatly accelerate materials matching to potential applications.

In this work, we present a multimodal model that utilizes data readily available at the point of MOF synthesis, specifically the powder X-ray diffraction (PXRD) pattern, represented as a spectrum, and the chemical precursors (metal and linker), encoded as text strings. To enhance the model's performance (particularly on small datasets), we leverage existing MOF structures from databases, represented as crystal graphs, to pretrain the model using a self-supervised learning framework. This pretraining enables the model to achieve high accuracy across a range of chemical, geometric, and quantum-chemical property predictions on low data regimes. We further evaluate the model's robustness under conditions of experimental structures or PXRD deviations from perfect crystals. Using this approach, we create a map, linking MOF synthesis to potential applications, offering a framework to recommend the best applications for the newly synthesized materials with minimal computational cost and complexity. Lastly, through a reverse time-travel study, we demonstrate that our method can identify materials suited for applications that are entirely different from those originally intended by their creators.

## Results

### Multimodal model development

To develop a predictive model for MOF properties, it is essential to encode both the material's chemistry and pore geometry. While this information can be extracted from the crystal structure of the MOF, crystal structure is typically not available at the synthesis stage. The synthesis of MOFs involves selecting precursors, namely the metal and organic linker, and finding the optimal synthesis conditions for the material to form. The primary characterization technique used to confirm successful MOF synthesis is a powder X-ray diffraction (PXRD) pattern. The PXRD pattern provides an abstract representation of the arrangement of atoms in the crystal, containing information about the material's global geometric structure. Previous studies have demonstrated the effectiveness of PXRD in capturing geometric information for machine learning predictions related to crystal system

classification, lattice parameters, and atomic positions[20–22]. Hence, in this work, we introduce a multimodal machine learning model that utilizes only the precursors and PXRD data as inputs–information readily available immediately following MOF synthesis (Fig. 1). The precursors provide valuable insights into the material's chemistry, which is supplemented by the PXRD, providing information about the global structure.

The chemical precursors of the MOF can be represented as a string. In this study, we utilize the Simplified Molecular Input Line Entry System (SMILES) to represent the organic linker, appending it with the type of metal used. We employ a transformer to embed this string, as previous research has shown the effectiveness of transformer architectures in property prediction, specifically using a string representation of the MOF[23,24]. Additionally, we utilize a convolutional neural network (CNN) to embed the PXRD pattern, which can be represented as a one-dimensional spectrum (see details in the Methods section).

While the precursors provide information on the chemistry of the metal and organic linker, they do not capture the metal-organic coordination environment, which can be necessary for predicting quantum chemical and chemistry-reliant properties. To inform the model about these coordination environments, we implement a self-supervised pretraining approach that leverages existing MOF crystal structures without the need for any labeled data. Our self-supervised pretraining combines a crystal graph convolutional neural network (CGCNN)[25] representation with our model. CGCNNs provide an expressive embedding of local chemical environments and have been successful in predicting quantum-mechanical properties of MOFs[18,19]. By pretraining our model with CGCNN, the embedding space learns meaningful representations related to properties such as electronegativity and covalent radius in local environments.

Finally, we utilize the pretrained model, which takes only PXRD and precursors, and fine-tune it with labeled data to predict desired properties. With the materials properties predicted, we use selection criteria for recommending them for various applications. Details of the method development can be found in SI and the method section.

### Model performance evaluation

We first demonstrate that our multimodal model effectively predicts a diverse range of MOF properties, laying the foundation of connecting MOF synthesis to applications. For our evaluation, we categorize the

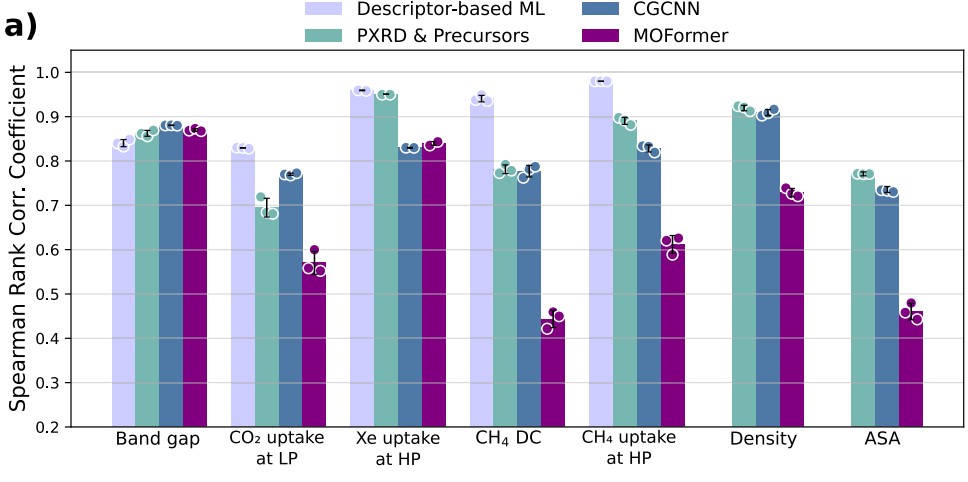

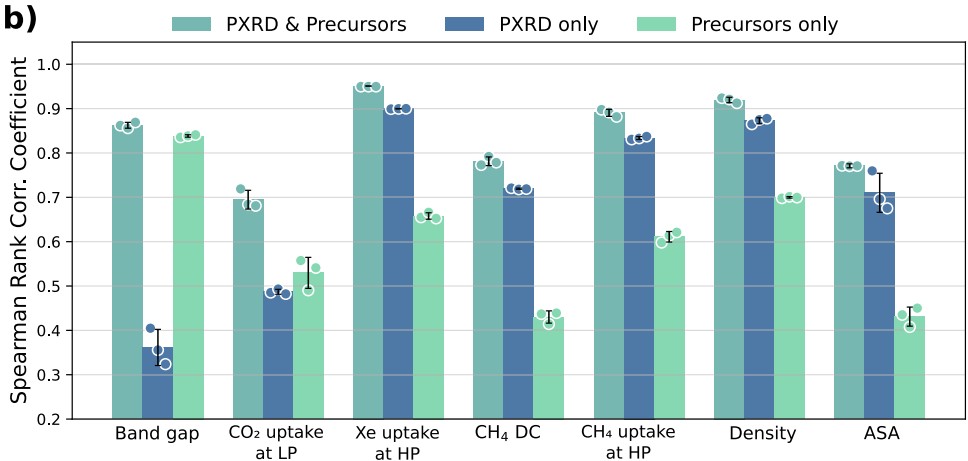

**Fig. 2 | Model performance across various property prediction tasks.**
**a** Regression results for our model (powder x-ray diffraction (PXRD) & precursors) in comparison to a descriptor-based machine learning (ML) model[26], a transformer-based model accepting MOFids (MOFormer)[23,24] and a crystal graph convolutional neural network (CGCNN, accepting 3D structures of crystals)[25] across various geometric ($CH_4$ deliverable capacity - DC, density, Xe uptake at HP - high pressure, $CH_4$ uptake at HP, accessible surface area - ASA), chemistry-reliant ($CO_2$ uptake at low pressure - LP) and quantum chemical (band gap) properties. **b** Ablation study showcasing the impact of multimodality across various properties. The results are reported in the mean Spearman's Rank Correlation Coefficient (SRCC), with the error bars showing standard deviation across three random splits. The results shown here are on structures from CoRE-2019[5,26] ($CO_2$ uptake at LP, $CH_4$ uptake at HP, ASA, density, $CH_4$ DC), QMOF[18,19] (band gap) and hMOF[26,29] (Xe uptake at HP) datasets. Source data are provided as a Source Data file.

properties into three groups: geometry-reliant, chemistry-reliant, and quantum-chemical properties. Geometry-reliant properties include pore geometric features such as accessible surface area (ASA) and properties including gas uptake at high pressures (HP) and methane deliverable capacity ($CH_4$ DC). Chemistry-reliant properties encompass gas uptake of $CO_2$ at low pressure (LP), while quantum-chemical properties focus on band gap. Data for these properties were extracted from prior studies and multiple databases, including QMOF, CoRE-2019, and the hMOF database (see SI for database details)[5,18,19,26–29].

Our multimodal model exhibits strong performance across the three property categories. Figure 2 quantifies the model's effectiveness in ranking materials using the Spearman rank correlation coefficient (SRCC) and accurate prediction of their properties using mean absolute error (MAE). Notably, its accuracy is comparable to machine learning models that utilize crystal structure data, as reported in prior studies[23,25,30,31]. Specifically, we benchmark our model against three state-of-the-art machine learning models: 1) Descriptor-based ML, which uses geometric and chemical descriptors (e.g., surface area, pore volume, and revised autocorrelation (RACs)[26]; see SI for details), 2) Crystal Graph Convolutional Neural Network (CGCNN)[25], which represents materials using a crystal graph, and 3) MOFormer[23], a model that takes a text-based representation of MOFs (MOFid[24]) as input.

While MOFormer uses a string representation, this encoding requires crystal structure information (e.g., MOF topology and metal cluster details), making it dependent on structural processing. Remarkably, our model performs comparably to crystal structure-based models across all property types. Specifically for geometric properties, such as accessible surface area (ASA), our model outperforms both CGCNN and MOFormer. The descriptor-based ML model achieves perfect scores for surface area and density, as these are directly provided as inputs, and therefore we left them out of Fig. 2. For geometry-reliant properties, such as high-pressure adsorption of $CH_4$ and Xe, our model achieves accuracy comparable to descriptor-based approaches. Finally, for chemistry-reliant and quantum-chemical properties, our model performs on par with crystal structure-based models (Fig. 2).

In scenarios where we deal with small labeled datasets, a self-supervised pretraining workflow can improve the model performance, where the pretraining leverages large amounts of unlabeled data to learn generalizable representations that can be fine-tuned on the limited labeled samples. In our study, we pretrain our multimodal model taking PXRDs and precursors against crystal structures available from large MOF databases. We show in the SI that this approach improves the performance of the model on small datasets, such as ARABG, and make the method applicable for the cases when large amount of data is

not available. However, it should be noted that the model performance has very negligible improvement when the pretrained model is fine-tuned on large MOF databases such as CoRE-MOF, BW20K, QMOF and hMOF.

It is insightful to compare the performance of our multimodal model with a model that relies solely on the chemical precursor and another model that relies solely on the PXRD. The results of this ablation study are shown on the right panel of Fig. 2. The results show that a model only accepting chemical precursors is inadequate for capturing the global structure of the MOF, scoring particularly low in geometric-reliant and pure geometric properties. On the other hand, a model only accepting PXRDs capture the global structure of the MOF well, but fails to capture the local environment, scoring low in chemistry-reliant and quantum chemical properties such as $CO_2$ uptake at LP and band gap. This confirms that the PXRD is critical and provides sufficient information about the geometry of a MOF, whereas the precursor aids in capturing information about the local environment. Achieving high accuracy in all three categories using only the information available at the time of MOF synthesis underscores the importance of multimodality in property prediction tasks. No single modality can fully encapsulate the complexities of chemistry and geometry in MOFs; thus, integrating both PXRD and precursor strings facilitates accurate prediction of material properties.

To further assess the model's generalizability, we assess its performance across various crystal systems. Analyzing the diversity within our dataset shows that model performance is not uniform across all crystal systems and depends on the database used for training and the specific crystal system of the MOF. Similar observations regarding the impact of crystal system diversity on model generalizability have been reported previously[26]. We found that training on more complex crystal systems, such as triclinic, leads to better generalization to less symmetric cases. Conversely, training on highly symmetric systems, such as cubic, results in poorer generalization. This suggests that crystal system diversity should be a consideration in database development and model evaluation, as well as the training of machine learning models. Further details regarding this analysis can be found in the Supplementary Information.

Given the high accuracy of our model in property prediction, we can extend its capabilities to build a recommendation system that connects MOF synthesis to applications. We note that while descriptor-based models offer slightly better performance in property prediction, they require preprocessed crystal structures in a computation-ready format (e.g., solvent removal and handling of partial occupancy), which can be challenging. In contrast, our approach leverages PXRD and precursor information, which are readily available upon synthesis, allowing us to provide accurate recommendations for both chemistry-reliant and geometry-reliant properties and applications without the need for extensive structural preprocessing.

## Robustness assessment

A strong indicator of the model's practicality is its evaluation against noisy or experimental data. The high performance of our multimodal model demonstrates that it is, in principle, feasible to use PXRD and precursors to accurately predict various properties of MOFs. Our model was trained using precursors and PXRD data derived from perfect, computation-ready crystal structures. While the precursors are devoid of inaccuracies due to the known composition prior to synthesis, experimentally measured PXRD patterns can be noisy and may differ from those simulated from ideal crystals. In the absence of large experimental PXRD databases, we evaluate the robustness of our method in two scenarios: first, using PXRDs computed from experimental crystal structures extracted from the CSD, and second, assessing the model's performance on some experimentally measured PXRDs. Given that PXRD primarily influences geometric-reliant properties, we expect minimal impact on predictions of chemistry-reliant

properties. Therefore, we focus our evaluation on a geometry-reliant property: predicting methane uptake at high pressure for methane storage applications[26].

Experimental crystal structures for MOFs often lack hydrogen atoms and may contain solvent molecules, partially occupied sites, or disordered atoms. To prepare these structures for computational chemistry calculations and property evaluations, certain adaptations are necessary. For example, to create a computation-ready format for gas adsorption calculations in the CoRE-2019 dataset, all solvents were removed. In the QMOF database for quantum calculations, hydrogen atoms were added, and unbounded solvents were eliminated. Consequently, the PXRD patterns computed from these modified structures differ from those available in the CSD. Figure 3 presents a parity plot illustrating differences in model predictions when using PXRDs computed from the CoRE-2019 and CSD, highlighting specific MOFs with three scenarios: missing hydrogen atoms and the presence of bounded/unbounded solvents. We observe that missing hydrogen atoms do not influence the PXRDs, and the model is very robust to these changes. This is due to the scattering power of a PXRD being proportional to the number of electrons in an atom, and hydrogen ($Z = 1$) atoms play a minimal role in scattering and the structural factor (see Methodology for more details). Furthermore, while the model generally shows robustness regarding bounded solvents, discrepancies can arise between predictions from computation-ready and CSD data. For instance, the MOF LELROL significantly deviates from parity[32]. However, we observe an overall robust predictions as the rankings of the MOFs in terms of $CH_4$ uptake is well captured (SRCC of 0.73) and a relative error below 13%. This demonstrates the model's robust performance in the presence of variations in crystal structures.

Up until now, the model's performance has been evaluated using simulated PXRDs. In practice, when assessing the crystallinity of MOF powders, experimental PXRDs may contain noise from instrumentation, temperature variations, and low-quality samples with poor crystallinity. To evaluate our model's robustness against these imperfections, we compare the recommendations for methane storage when using simulated versus experimental PXRDs. This evaluation was conducted on five materials, for which both experimental and simulated PXRDs are shown in Fig. 4. In general, we observe consistent recommendations between the simulated and experimental PXRDs, except for one case. In this inconsistent case, the difference between the simulated and experimental PXRDs is pronounced; the experimental PXRD exhibits noticeable noise and multiple significant peaks compared to the simulated PXRD. Additional comparisons can be found in the Supplementary Information, alongside model assessment on different noise levels. Overall, our evaluation indicates that the model remains robust to a great extent in the presence of noise and imperfections in PXRDs. This enables experimentalists to use their measured PXRDs to predict their properties.

## An application recommendation map

The high performance of our multimodal model in predicting various properties of MOFs enables a direct link between their synthesis and potential applications. We built a recommendation system that classifies MOFs as promising candidates for specific applications based on predicted material properties. The applications explored in this study include gas storage (for gases such as methane, xenon, and hydrogen) and carbon capture, including direct air capture (DAC). The metrics for these applications are derived from industrial cases reported in the literature (see SI for details)[33–37]. Our results show accurate distinction of promising MOFs for these applications (see SI for details). To minimize false negatives—instances where promising candidates are overlooked, we configure our model to be more optimistic in its recommendations.

For large-scale application predictions, such as assessing all materials in the CoRE-2019, hMOF and QMOF databases, it is

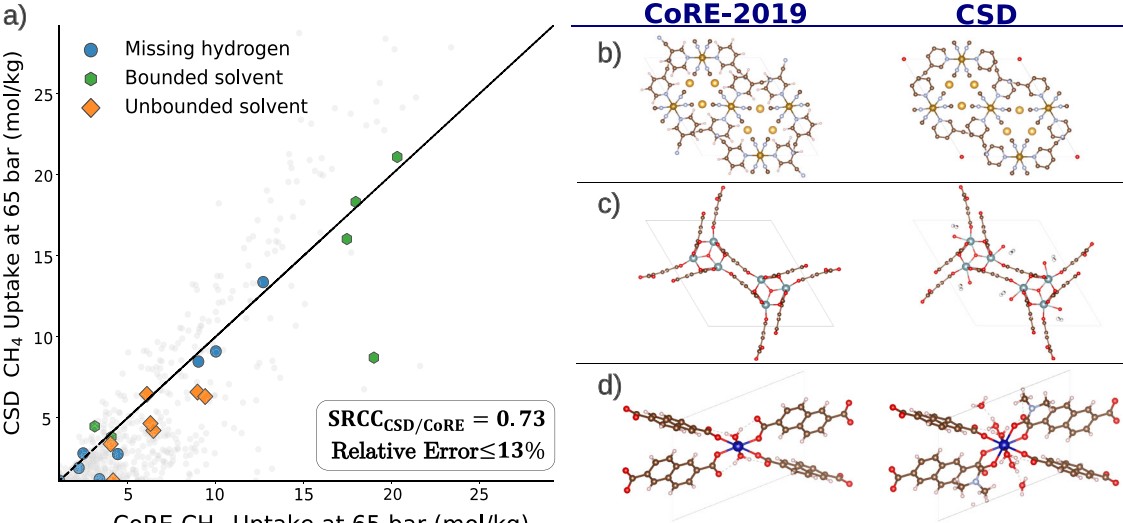

**Fig. 3 | Assessment of robustness of model on the crystal structure imperfections.** The parity plot in (**a**) shows a comparison between the model output when fed a powder x-ray diffraction (PXRD) pattern computed from computation-ready crystal structure (CoRE-2019)[5] and the experimental crystal structure from the Cambridge Structural Database (CSD)[7]. The parity plot showcases a set of selected examples of metal-organic frameworks (MOFs) that either have hydrogen atoms (blue), MOFs with bounded solvents present (green) and MOFs with unbounded solvents (yellow), with the rest of the MOFs grayed out as they are not highlighted examples. On the right panels, we showcase examples of MOFs within these three classes, with the left column being CoRE-2019 and right column being CSD structure. **b** Showcases a MOF (CSD reference code: WAHMEY)[56] (metal types: Fe, Au, linker: $C_6H_4N_2$) with missing hydrogen atoms, **c** showcases a MOF (CSD reference code: NALYEG)[57] (metal type: U, linker: $C_9H_6O_6$) with unbounded solvent in the form of $C_3H_7NO$, **d** showcases a MOF (CSD reference code: VEFLUP)[58] (metal type: Co, linker: $C_{12}H_8O_4$) with bounded solvent in the form of $C_3H_7NO$. Crystal structures visualized using VESTA[59] and the atom color coding is: N (blue), O (red), H (white), C (brown), Fe (amber), Au (light amber), U (gray), Co (dark blue). Source data are provided as a Source Data file.

insightful to see how the model links synthesis information to applications. Figure 5 shows the projection of the model's embedding space for materials in CoRE-2019, with promising MOFs color-coded for various applications. The visual trends highlight the model's ability to identify application-relevant properties from the latent space, effectively clustering MOFs based on their suitability for specific applications. Notably, MOFs flagged as promising for gas storage are distinctly clustered on the left side of the map. Further analysis of the map, color-coded by material density and pore diameter, indicates that these materials are indeed more porous and have lower densities–attributes favorable for gas storage applications, as supported by previous studies[26,38]. Additionally, the map provides insights into the desired metal electronegativity when designing materials for carbon capture or DAC. These insights confirm the relation between the embedding space of the multimodal model to chemical and structural attributes of MOFs. The model's ability to infer these structural attributes solely from PXRD and precursor data is a significant strength. In the previous section, we evaluated the model's performance against experimental imperfections and found that model confidence correlates with agreement to experimentally determined structures. This capability enables effective materials-application matching for newly discovered MOFs.

The map shown in Fig. 5 provides insights into the designability of MOFs for various applications. Overlaps between regions for specific applications suggest the potential for versatile MOFs that can serve multiple purposes, while shedding light on the properties most relevant for designing these MOFs. For instance, many MOFs flagged as promising for methane storage are also recommended for xenon and hydrogen storage, indicating relationships between materials and applications. Moreover, the map illustrates that for a particular application, there can be different families of MOFs that perform optimally. For instance, there are a few different clusters showing promising performance for carbon capture, balancing pore structure with specific chemical interactions.

## Discussion

Our study offers a pathway for matching newly discovered materials to their potential applications. This was achieved by developing a multimodal model that requires only data available upon MOF synthesis — namely PXRD and chemical precursors, enabling accurate predictions of a wide range of geometry-reliant, chemistry-reliant, and quantum-chemical properties for MOFs. This remarkable accuracy on such a wide range of properties underscores the importance of leveraging multimodality in materials property prediction, as different modalities enhance the model's ability to predict a broader range of properties, which a single modality cannot achieve.

A practical motivation for our method was to identify applications for newly discovered MOFs, which may differ significantly from the intended application at the time of design. To demonstrate this capability, we conducted a time-travel experiment, training our model on CoRE-2019 entries deposited in the CSD database before 2017, with a test set comprising entries deposited after 2017. This approach assesses whether our model can recommend applications for MOFs synthesized in the future. Our recommendation system identified 18 promising candidates for carbon capture, with an impressive accuracy of 16 out of 18 based on molecular simulation data. Remarkably, 15 of these materials were originally designed for applications other than $CO_2$ capture. Figure 6 presents some of these MOFs along with their intended synthesis applications based on the corresponding articles. Please refer to the Supplementary Information for the results on the rest of the MOFs.

Given that our model can recommend applications based solely on a PXRD pattern and precursors, experimentalists can utilize our web app to receive recommendations immediately after conducting crystallinity assessments to obtain PXRD data[39]. By integrating advanced key performance indicators from chemical process modeling and life-cycle assessments[40] as well as robotic platforms[41], this approach will accelerate materials discovery for sustainable applications.

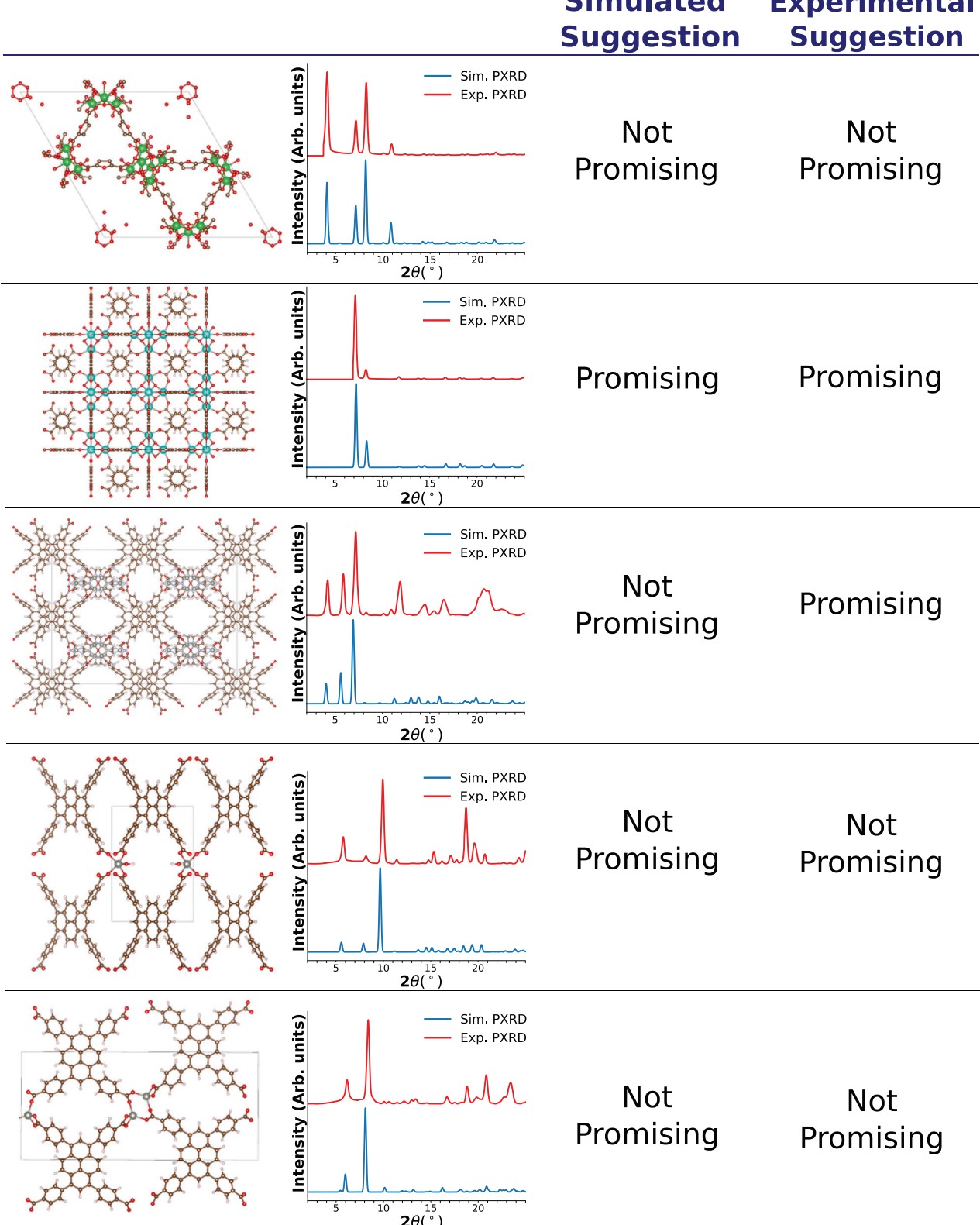

**Fig. 4 | Comparison of recommendations made by model with simulated and experimental powder x-ray diffraction (PXRD) patterns.** This assessment was done on methane storage recommendations due to the dependence on pore volume and density. The blue and red PXRDs correspond to simulated and experimental PXRDs respectively. The table showcases results (with metal (M) and linker (L)) for: CAU-28 (M: Zr, L: $C_6H_4O_5$), Yb-UiO-66 (M: Yb, L: $C_8H_6O_4$), Zn-(Ade)(TBAPy) (M: Zn, L: $C_5H_5N_5$, $C_{44}H_{26}O_8$), $Zn_2$-(TBAPy) (M: Zn, L: $C_{44}H_{26}O_8$) with coordinated water molecules and with a distorted structural framework[60–62]. Crystal structures visualized using VESTA[59] and the atom color coding is: N (blue), O (red), H (white), C (brown), Zr (green), Yb (light blue), Zn (gray).

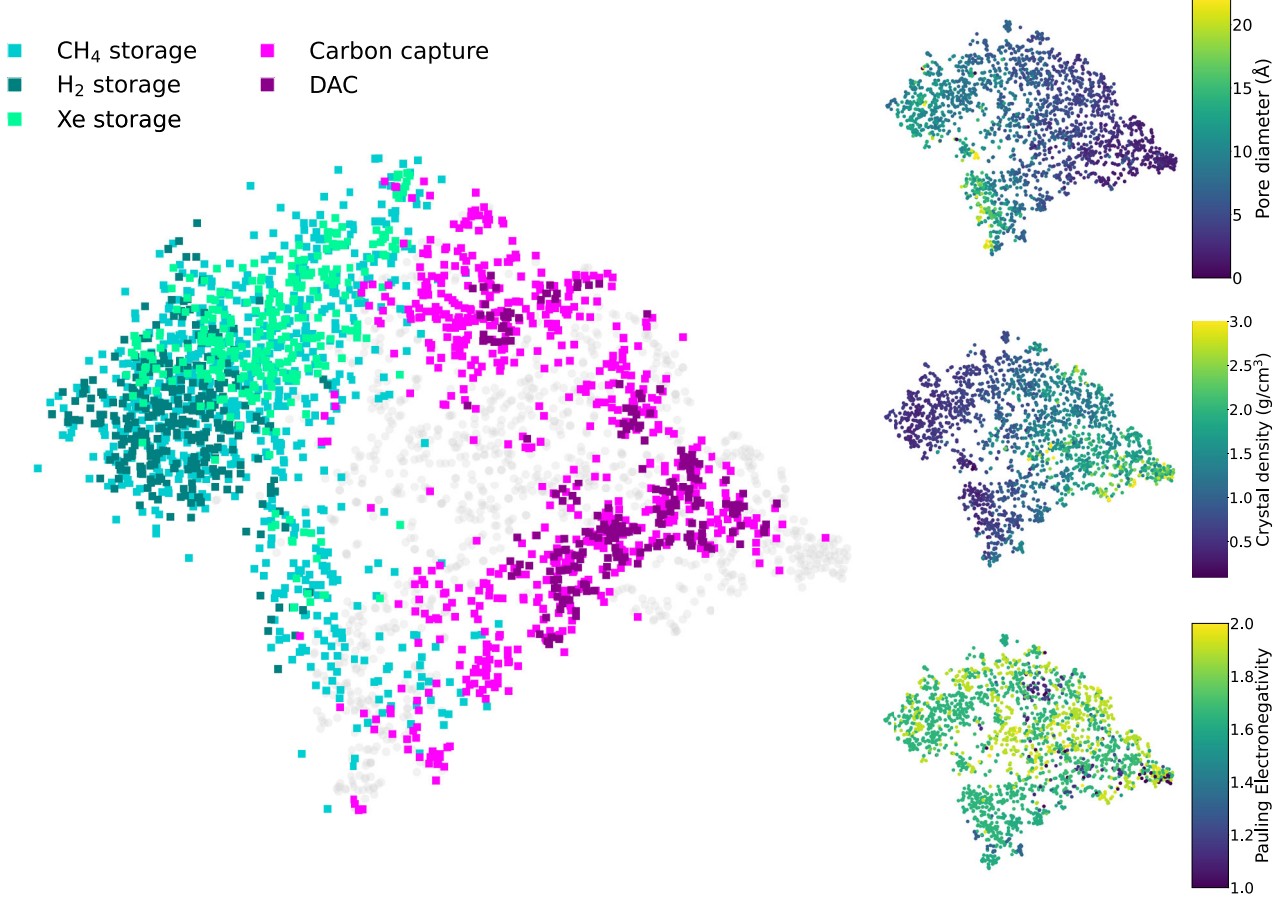

**Fig. 5 | Mapping synthesis to applications.** The projection of the latent space of the multimodal model is obtained using t-SNE (t-distributed Stochastic Neighbor Embedding). The different colors indicate the model's recommended applications for various metal-organic frameworks (MOFs) in the combined CoRE-2019, QMOF and hMOF databases[5,18,19,26,29]. The panels on the right display the same embedding space color coded with specific physiochemical properties: crystal density (top right), pore diameter (middle right) and Pauling electronegativity (bottom right). Source data are provided as a Source Data file.

## Methods

### Generating input to the model

From leveraging the crystal structures from each database (CoRE-2019, BW20K, ARABG, QMOF, hMOF)[5,18,19,28,29], pymatgen's XRD module[42] was used to calculate the simulated PXRD pattern for every MOF specified, ranging from 0 to 90 degrees. It should be noted that for pymatgen, the Debye-Waller factor (for temperature) and the multiplicity factor are not accounted for. Furthermore, CuKa radiation was used, with no refinement on the structures done[42,43]. Please refer to the SI for more details regarding the generation of simulated PXRD patterns.

When a PXRD is collected, there are usually a varying number of data points. When machine learning is done, the inputs need to all be of the same length. In order to construct a 1D array of intensities, interpolation can be done to fill in the missing data points. It should be noted that experimental PXRDs characteristically have broad, smooth peaks due to various reasons such as instrumentation. In an attempt to model this behavior, a Gaussian transformation was done on the 1D array. The total desired size of the 1D PXRD vector is 9000, with the desired peak width being controlled by $\sigma = 0.1$. For details regarding the configurations, parameters and algorithm of the preprocessing and transformation of the PXRD pattern, please refer to the SI.

The metal nodes and the SMILES of the organic linker were previously given for MOFs across various databases[5,18,19,28,29]. The chemical precursors were constructed in the format of: [Metal Type].[Organic Linker]. This was inputted and tokenized by the transformer channel of our model.

### Construction of model

Taking inspiration from MOFormer, an encoder and tokenizer was built on top of the transformer[23]. The precursor is inputted into a SMILES tokenizer (as the only portion inputted from the precursor are the inorganic and organic building blocks), with a [CLS] and [SEP] tag added at the beginning and the end of the precursor string respectively. As the precursor strings are of varying lengths, there is padding added to each string with [PAD] such that they are at a fixed length of 512 characters. Positional encoding is done on the tokens to give information on the absolute and relative positions of each token, where the dimension of embedding is 512. The transformer is built on a self-attention mechanism[44]. The encoder layers consist of multihead attention layers, where each encoder has the attention layer, followed by the multilayer perceptron (MLP) layer. In each attention layer, the input sequence is broken down into three vectors: the query (Q), key (K) and value (V), with the corresponding learnable weights being $W_Q$, $W_K$, $W_V$ respectively. The attention output is defined as:

$$A = \text{softmax}\left(\frac{QK^\top}{\sqrt{d_k}}\right)V \qquad (1)$$

Based off this, the model is able to focus on different parts of the inputted precursor. The total number of tokens in the vocabulary is 4021, with the number of parallel heads being 8 and the number of encoder layers being 6. A dropout rate of 0.1 was used. An embedding is returned of shape (N, 512, 512), where N is the batch size (32 is chosen for fine-tuning). Chitturi et al. published work in which they built a 1D-

| | Intended Application | CO₂ Uptake at 0.15 bar (mol/kg) |
|---|---|---|
| EXUHUC | Thermal Stability | 4.66 |
| GARLUJ | Luminescent Material | 3.53 |
| LAQZOV | Fluoride Adsorption | 5.70 |
| NAMDUD | Carbon Dioxide Adsorption | 3.68 |
| NAWKII | Catalyst | 6.29 |

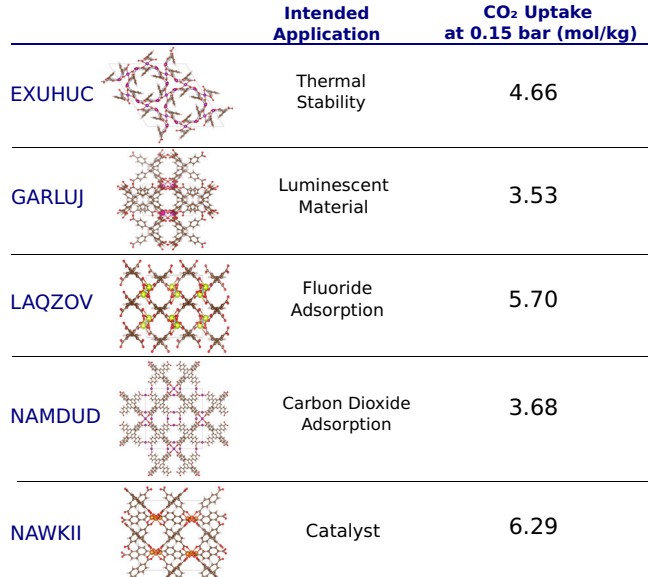

**Fig. 6 | A "time-travel" experiment connecting MOFs and their potential applications.** Metal-organic frameworks (MOFs) identified to be promising for CO₂ capture (from their predicted computational CO₂ uptake at 0.15 bar) and the intended applications reported in their original articles. The MOFs, reported from top-to-bottom in their Cambridge Structural Database (CSD) reference codes, are EXUHUC (metal type: Mn, linker: $C_{11}H_8O_4N_2$)[63], GARLUJ (metal type: Eu, linker: $C_{15}H_{13}NO_4$)[64], LAQZOV (metal type: Ce, linker: $C_{21}H_{14}O_6$)[65], NAMDUD (metal type: Mn, linker: $C_{27}H_{18}O_6$)[66] and NAWKII (metal type: Mg, linker: $C_{12}H_8O_4$)[67].

CNN that accepts PXRD patterns as the input for the prediction of the classification of crystal symmetries[45]. Using this as inspiration, a 1D-CNN was constructed with ten convolutional blocks, with max-pooling layers and ReLU activation functions following each convolutional block. After the 10th convolutional block, the embedding of the CNN is returned of shape (N, 660). Taking the two embeddings from the transformer and CNN, tensor concatenation was performed to return a concatenated embedding size of (N, 1172). This concatenated tensor is fed into a projector, in which linear and softplus activation functions are used to project a tensor of shape (N, 512). This embedding is used in the self-supervised pretraining against the crystal graph convolutional neural network (CGCNN). When performing regression, the projected embedding is fed into a regression head consisting of linear and ReLU Dense layers to return M labels, with M being any number of desired properties the user wants to return. For the property prediction results, M was set to 1 (one label was predicted at a time). PyTorch 2.3 has been used in the construction of the model[46,47].

For model evaluation, data visualization and analysis, we used Pandas[48], Matplotlib[49], NumPy[50] and Scikit-learn[51].

### Self-supervised learning

As DFT and experimental labels for MOFs are difficult to obtain, it is difficult to construct large databases for machine learning applications. As a result, there is an incentive to construct models which perform well in lower data regimes. Furthermore, while the PXRD gives adequate information on the global environment of the MOF, capturing the metal/organic chemistry of MOFs by using a PXRD/precursors is difficult due to the model not having an intuitive understanding of the chemistry behind the tokens of a text representation of a MOF. Self-supervised pretraining aims to solve this problem, as a pretrained model's weights are initialized such that it converges to a solution quickly, rather than using randomized weights in which large amounts of data is needed. To avoid the limitation of the model not understanding the local environment of a MOF from the

inputs, a self-supervised learning (SSL) pipeline has been constructed where representation learning is done between a crystal graph convolutional neural network (CGCNN) and our model[23,52]. The CGCNN accepts a 3D structure of a MOF and returns an embedding containing information on the structure itself, chemical composition and local environment. While the geometric properties, which are reliant on structural information, are already easy to predict due to the PXRD input, chemistry-reliant properties are harder due to the inputs not giving the model an explicit understanding of the metal/organic chemistry of the MOF. This gives us an incentive to perform self-supervised learning against the CGCNN embeddings. The embeddings are taken out of the projector for both the CGCNN and our model, each with an embedding size of 512. From these two embeddings, it is possible to construct a cross-correlation matrix of shape (512, 512). This can be defined as:

$$C_{ij} \overset{\Delta}{=} \frac{\sum_b z_{b,i}^A z_{b,j}^B}{\left(\sqrt{\sum_b z_{b,i}^A}\right)\left(\sqrt{\sum_b z_{b,j}^B}\right)} \quad (2)$$

Where b is the batch sample, i, j are the indices of the vector dimension, $z^A$, $z^B$ are the embeddings for our model and CGCNN respectively. For representation learning to happen, the cross-correlation matrix should approach an identity matrix. As a result, the loss function (Barlow-Twin loss) used to quantify this can be defined as:

$$L_{BT} \overset{\Delta}{=} \sum_i (1 - C_{ii})^2 + \lambda \sum_i \sum_{j \neq i} C_{ij}^2 \quad (3)$$

Where $\lambda$ is the regularization term which describes the trade-off between the first and second parts of the loss function[53]. A batch size of 64 was used, with the regularization term being 0.0051. Furthermore, the learning rates of CGCNN and our model is 0.00005 for 100 epoch.

### Supplementary information availability

Information about the MOF databases used, model architecture/hyperparameters, relevant property distributions, CoRE-2019 filtration process, statistics of model performance, recommendation system details (thresholds, masked learning), the "time-travel" model discussed in the MOF Application Discovery section, evaluation on CoRE-2019's diversity, model robustness on experimental PXRDs and its robustness on covalent-organic frameworks (COFs) can be found in the supplementary information section. Data efficiency information can also be found in the supplementary information.

### Reporting summary

Further information on research design is available in the Nature Portfolio Reporting Summary linked to this article.

## Data availability

The powder x-ray diffraction, precursor, and the labels used from the databases showcased in this study have been deposited in the Zenodo repository with this link: https://zenodo.org/records/14908210. Source data for all figures and tables are provided as a Source Data file.

## Code availability

All code related to this study was developed in Python and is publicly available under the MIT License, an open-source license approved by the Open Source Initiative. The full source code (including the featurization, prediction and trained models) are available from https://github.com/AI4ChemS/XRayPro[54].

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

## Acknowledgements

The authors gratefully acknowledge financial support from Natural Sciences and Engineering Research Council of Canada, the University of Toronto's Acceleration Consortium through the Canada First Research Excellence Fund under Grant number CFREF-2022-00042, and National Research Council of Canada under the Materials for Clean Fuels Challenge Program grant number MCF-146. We would like to thank Hudson Bicalho and Dr. Micaela Richezzi from Dr. Ashlee Howarth's group (Concordia University) for providing us experimental PXRDs for materials in Fig. 4.

## Author contributions

Conceptualization: S.T.K. & S.M.M.; Data curation: S.T.K.; Formal analysis: S.T.K.; Funding acquisition: S.M.M.; Investigation: S.T.K. & S.M.M.; Methodology: S.T.K. & S.M.M.; Project administration: S.M.M.; Software: S.T.K.; Supervision: S.M.M.; Validation: S.T.K.; Visualization: S.T.K.; Writing - review & editing: S.T.K. & S.M.M.

## Competing interests

The authors declare no competing interests.
