## [Peer Review file · Nature Communications]

Connecting metal-organic framework synthesis to applications with multimodal machine learning

Corresponding Author: Dr Seyed Mohamad Moosavi

Version 0:

Reviewer comments:

Reviewer #1

(Remarks to the Author)

Report on "Connecting metal-organic framework synthesis to applications with a self-supervised multimodal model", corresponding author: Seyed Mohamad Moosavi

- What are the noteworthy results?

The study presents an innovative synthesis-to-application recommendation system for MOFs, using only PXRD data and MOF building blocks as input. It provides an easy access web-app and jupyter notebook tutorial that lower the barrier of entry and as the potential to reach a broad audience.

- Will the work be of significance to the field and related fields? How does it compare to the established literature? If the work is not original, please provide relevant references.

Yes, the approach is original and addresses a key challenge in MOFs.

- Does the work support the conclusions and claims, or is additional evidence needed?

Yes but it lacks some metrics and comparison with experimental data.

- Are there any flaws in the data analysis, interpretation and conclusions? - Do these prohibit publication or require revision?

No major flaws are apparent but slight additional work is needed.

- Is the methodology sound? Does the work meet the expected standards in your field?

The methodology appears sound and aligns with current practices in the field.

- Is there enough detail provided in the methods for the work to be reproduced?

The absence of a table with the direct prediction of the model and metrics other than Spearman coefficient makes it difficult to reproduce. Same for the PXRD, if ones want to use an experimental PXRD the conversion process is not straightforward and might hinder future reproducibility efforts.

Comment to the author:

We acknowledge the reading quality of the article and to original methodology and approach. We recommend the article to Nature Communications with minor revisions; see comments above.

Reviewed by: Jennifer T Chayes (senior reviewer) and Theo Jaffrelot (junior reviewer)

(Remarks on code availability)

Reviewer #2

(Remarks to the Author)

This paper presents a self-supervised multimodal model for predicting the properties and applications of metal-organic frameworks (MOFs) based on synthesis data, including powder X-ray diffraction (PXRD) patterns and chemical precursors. The authors demonstrate the model's effectiveness in generating property predictions and making application recommendations, including for future MOF discoveries. The manuscript provides a solid approach and promising results, but there are aspects that need further clarification and analysis to strengthen the findings.

My concerns about the paper are as follows:

1. Effectiveness of Self-Supervised Learning

The loss curves for self-supervised training (flattening out at 40 epochs), combined with the small size of the pretraining dataset, make it unclear whether the approach is genuinely beneficial. While the authors report improved MAE for the pretrained models compared to training from scratch, the absence of standard deviations makes it hard to assess the reliability of these results. Were the test sets randomized? Was any k-fold cross-validation conducted? Is the improvement statistically significant? Pretraining typically demonstrates greater benefits on larger datasets, as shown in prior works, so it is not clear if the approach adds significant value here.

2. Clarity of Results in Figure 2

Figure 2 is slightly difficult to interpret as presented. Including a table to accompany the figure would make the results easier to understand and improve clarity.

3. Descriptor-Based Models

Descriptor-based models perform better on most tasks in Figure 2. Have these descriptors been used in MOF recommendation systems? Would they give more accurate results given their good performance on tasks in Figure 2?

4. "Time-Travel" Generalization

The "time-travel" experiment is promising but raises questions about the similarity between the training and test sets. How distinct are the MOFs in the test set compared to those in the training set? Are the results reflective of the model's true generalizability, or is it leveraging patterns from previously seen data? An analysis of embedding similarity between test points and their closest training samples could help clarify this.

5. Threshold Criteria

The justification for specific thresholds in the recommendation system (e.g., methane uptake values) is missing. Providing reasoning for these thresholds in the SI would improve the clarity of the recommendations.

6. Dependence on PXRD Quality

While robustness testing is included, the impact of significant noise or distortions in PXRD data is not fully explored. For example, the authors state, "In this inconsistent case, the difference between the simulated and experimental PXRDs is pronounced; the experimental PXRD exhibits noticeable noise and multiple significant peaks compared to the simulated PXRD." How is "pronounced" defined in this context? Quantifying these differences or providing clear metrics would improve clarity and allow for a better assessment of the model's performance under noisy or distorted conditions.

7. Baseline Comparisons

There is a lack of comparison to similar baselines on property prediction benchmarks, such as MOFormer or CGCNN.

(Remarks on code availability)

Reviewer #3

(Remarks to the Author)

(Remarks on code availability)

The code is well-written and accessible to both researchers with strong and limited programming experience. However, to further improve accessibility for a broader audience and enhance reproducibility, the authors should:

- Provide a script to directly load the model and generate predictions.
- Add a tutorial on converting experimental PXRD data into a format compatible with the model.
- Modify the tutorial to include the recommended Python version, as it currently fails to work if the Python version is not specified.
- Clarify how uploaded data on the web app is handled—whether it is stored in an external database or removed after a certain period—this is especially critical for newly synthesized MOFs or data from companies.

Version 1:

Reviewer comments:

Reviewer #2

(Remarks to the Author)

Comments to the authors:

1. In the main section authors mention: "This pretraining enables the model to achieve high accuracy across a range of chemical, geometric, and quantum-chemical property predictions." However, in their response letter they have stated: "However, we acknowledge that for larger datasets (e.g., CoRE-MOF, BW20K, hMOF, and QMOF), the availability of sufficient training data makes training from scratch generally effective." This limitation should be explicitly acknowledged in the main text, as self-supervised learning shows no tangible benefit for larger datasets. I would suggest that the authors also acknowledge this limitation in the main section or any other section of the manuscript.

2. The authors have addressed all my other concerns

(Remarks on code availability)

I did not run the code but the repo has instructions for installation and running the code

Reviewer #3

(Remarks to the Author)

The authors have addressed our comments from the last round satisfactorily. We strongly recommend publication in Nature Communications.

(Remarks on code availability)

The code is user-friendly and easy to download. They also provided a tutorial that help reproduce some of the results of the paper. They also have a web application for those that have issue with programming that allow a to target a broader audience.

Response to Reviewers for “Connecting metal-organic framework synthesis to applications with multimodal machine learning”

We thank the reviewers for their valuable comments and constructive feedback, which have helped us improve the quality of our manuscript; we have addressed each point in detail below. The reviewers’ comments are in gray, the authors’ response in blue and changes in the manuscript in green.

Reviewer #1 Comments

Report on “Connecting metal-organic framework synthesis to applications with a self-supervised multimodal model”, corresponding author: Seyed Mohamad Moosavi

Reviewer Comment: - What are the noteworthy results? The study presents an innovative synthesis-to-application recommendation system for MOFs, using only PXRD data and MOF building blocks as input. It provides an easy access web-app and jupyter notebook tutorial that lower the barrier of entry and as the potential to reach a broad audience.

Reviewer Comment: - Will the work be of significance to the field and related fields? How does it compare to the established literature? If the work is not original, please provide relevant references. Yes, the approach is original and addresses a key challenge in MOFs.

Reviewer Comment: - Does the work support the conclusions and claims, or is additional evidence needed? Yes but it lacks some metrics and comparison with experimental data.

Authors Response: We agree with the reviewer on these points. We include additional metrics, namely mean absolute error (MAE), mean squared error (MSE) and root mean squared error (RMSE), alongside the spearman rank correlation coefficient (SRCC) that was shown throughout the paper. The complete set of metrics are included in the revised Supplementary Information for the regression results across different properties and benchmark datasets and models (see full update in the comments below).

Reviewer Comment: - Are there any flaws in the data analysis, interpretation and conclusions? - Do these prohibit publication or require revision? No major flaws are apparent but slight additional work is needed.

Reviewer Comment: - Is the methodology sound? Does the work meet the expected standards in your field? The methodology appears sound and aligns with current practices in the field.

Reviewer Comment: - Is there enough detail provided in the methods for the work to be reproduced? The absence of a table with the direct prediction of the model and metrics other than Spearman coefficient makes it difficult to reproduce. Same for the PXRD, if ones want to use an experimental PXRD the conversion process is not straightforward and might hinder future reproducibility efforts.

Authors Response: We agree with the reviewer comments. In the revised manuscript, we have explicitly included the regression results in both the main text (Figure 2) and the Supplementary

Information in tabular format to enhance reproducibility. In the updated Figure 2, we first quantitatively compare our model against state-of-the-art models such as CGCNN and MOFormer (left panel, radar plot). In addition, we conduct an ablation study to demonstrate the necessity of incorporating both PXRD and precursor information to achieve high model performance.

We also acknowledge the reviewer’s concern regarding the reproducibility of PXRD pre-processing. To improve clarity, we have created a public repository containing the computed PXRDs for each database along with a Python script for pre-processing the data. Additionally, we provide iPython notebooks as a step-by-step guide for utilizing the dataset, available at: iPython notebooks Further details on the revised figure, code availability, and benchmark regression results are provided below.

Figure 2: Model performance across various property prediction tasks. Left panel: Regression results for our model (PXRD & precursors) in comparison to a descriptor-based model, a transformer-based model accepting MOFids (MOFormer) and a crystal graph convolutional neural network (CGCNN, accepting 3D structures of crystals) across various geometric, chemistry-reliant and quantum chemical properties. Right panel: Ablation study showcasing the impact of multimodality across various properties. The results are reported in Spearman’s Rank Correlation Coefficient (SRCC). The results shown here are on CoRE-2019 (CO₂ uptake at LP, CH₄ uptake at HP, surface area, density, CH₄ deliverable capacity), QMOF (band gap) and hMOF (Xe uptake at HP) datasets.

“Code Availability The featurisation, prediction, and trained models are available from <https://github.com/AI4ChemS/XRayPro>. ”

“SI Table S9 Performance metrics for crystal graph convolutional neural network (CGCNN). The units for gas uptake, band gap, H₂ capacity, pore diameter, ASA and density are mol/kg, eV, g/L, Angstrom, m²/cm³ and g/cm³ respectively. This was run over 3 random seeds.”

Property	MAE	MSE	RMSE	SRCC
CH ₄ uptake at HP	1.52 ± 0.04	4.17 ± 0.26	2.04 ± 0.06	0.82 ± 0.01
CO ₂ uptake at LP	0.73 ± 0.006	1.02 ± 0.05	1.01 ± 0.03	0.77 ± 0.002
Band gap	0.34 ± 0.009	0.22 ± 0.006	0.46 ± 0.006	0.88 ± 0.001
H ₂ capacity	2.51 ± 0.08	11.95 ± 0.19	3.45 ± 0.03	0.92 ± 0.006
Xe uptake at HP	1.02 ± 0.02	1.72 ± 0.008	1.31 ± 0.003	0.83 ± 0.001
Pore diameter	1.68 ± 0.01	6.74 ± 0.07	2.59 ± 0.01	0.61 ± 0.01
ASA	410.4 ± 11.8	283966 ± 9459	532 ± 8.8	0.74 ± 0.01
Density	0.12 ± 0.004	0.03 ± 0.002	0.16 ± 0.01	0.91 ± 0.01
CH ₄ DC	25.5 ± 0.46	1103 ± 46.4	33.2 ± 0.69	0.77 ± 0.01

“SI Table S10 Performance metrics for MOFormer. The units for gas uptake, band gap, H₂ capacity, pore diameter, ASA and density are mol/kg, eV, g/L, Angstrom, m²/cm³ and g/cm³ respectively. This was run over 3 random seeds.”

Property	MAE	MSE	RMSE	SRCC
CH ₄ uptake at HP	2.47 ± 0.08	12.72 ± 1.28	3.56 ± 0.18	0.63 ± 0.02
CO ₂ uptake at LP	0.95 ± 0.01	1.78 ± 0.01	1.33 ± 0.01	0.54 ± 0.00
Band gap	0.36 ± 0.003	0.25 ± 0.003	0.50 ± 0.003	0.87 ± 0.003
H ₂ capacity	3.04 ± 0.07	21.55 ± 1.49	4.63 ± 0.16	0.90 ± 0.01
Xe uptake at HP	0.92 ± 0.002	1.70 ± 0.002	1.30 ± 0.00	0.84 ± 0.005
Pore diameter	1.75 ± 0.04	7.27 ± 0.58	2.69 ± 0.11	0.49 ± 0.02
ASA	574.1 ± 18.3	574021 ± 55376	756.7 ± 36.1	0.46 ± 0.05
Density	0.22 ± 0.01	0.09 ± 0.001	0.30 ± 0.002	0.73 ± 0.01
CH ₄ DC	36.53 ± 0.86	2291.5 ± 183.9	47.8 ± 1.94	0.50 ± 0.02

“SI Table S11 Performance metrics for an XGBoost model accepting RACs and geometric descriptors. The units for gas uptake, band gap, H₂ capacity, pore diameter, ASA, and density are mol/kg, eV, g/L, Å, m²/cm³, and g/cm³, respectively. This was run over 3 random seeds.”

Property	MAE	MSE	RMSE	SRCC
CH ₄ uptake at HP	0.45 ± 0.01	0.55 ± 0.12	0.73 ± 0.08	0.98 ± 0.00
CO ₂ uptake at LP	0.54 ± 0.01	0.65 ± 0.03	0.81 ± 0.02	0.83 ± 0.001
Band gap	0.41 ± 0.03	0.38 ± 0.04	0.62 ± 0.03	0.84 ± 0.009
H ₂ capacity	0.76 ± 0.01	1.15 ± 0.01	1.07 ± 0.003	0.99 ± 0.00
Xe uptake at HP	0.45 ± 0.004	0.42 ± 0.01	0.65 ± 0.008	0.96 ± 0.0008
Pore diameter	0.045 ± 0.01	0.24 ± 0.15	0.46 ± 0.17	0.99 ± 0.00
ASA	2.87 ± 0.15	59.09 ± 22.71	7.54 ± 1.49	0.97 ± 0.002
Density	0.0042 ± 0.00	0.00036 ± 0.00	0.02 ± 0.004	0.99 ± 0.00
CH ₄ DC	10.62 ± 0.27	245.1 ± 15.6	15.64 ± 0.51	0.94 ± 0.004

“SI Table S12 Performance metrics for transformer only accepting precursors. The units for gas uptake, band gap, H₂ capacity, pore diameter, ASA, and density are mol/kg, eV, g/L, Å, m²/cm³, and g/cm³, respectively. This was run over 3 random seeds.”

Property	MAE	MSE	RMSE	SRCC
CH ₄ uptake at HP	2.53 ± 0.11	13.37 ± 1.30	3.65 ± 0.17	0.60 ± 0.02
CO ₂ uptake at LP	1.00 ± 0.03	1.84 ± 0.07	1.35 ± 0.03	0.53 ± 0.03
Band gap	0.41 ± 0.003	0.31 ± 0.002	0.56 ± 0.002	0.84 ± 0.002
H ₂ capacity	7.46 ± 0.002	86.87 ± 1.72	9.32 ± 0.09	0.47 ± 0.002
Xe uptake at HP	1.44 ± 0.02	3.27 ± 0.02	1.81 ± 0.02	0.67 ± 0.01
Pore diameter	1.99 ± 0.03	10.41 ± 0.93	3.22 ± 0.14	0.39 ± 0.03
ASA	604.60 ± 3.72	545711 ± 10513	738.6 ± 7.10	0.41 ± 0.03
Density	0.23 ± 0.002	0.09 ± 0.002	0.31 ± 0.002	0.70 ± 0.002
CH ₄ DC	38.4 ± 1.29	2488.5 ± 2.23	49.88 ± 0.02	0.47 ± 0.02

“SI Table S13 Performance metrics for CNN only accepting PXRD. The units for gas uptake, band gap, H₂ capacity, pore diameter, ASA, and density are mol/kg, eV, g/L, Å, m²/cm³, and g/cm³, respectively. This was run over 3 random seeds.”

Property	MAE	MSE	RMSE	SRCC
CH ₄ uptake at HP	1.72 ± 0.04	5.96 ± 0.49	2.44 ± 0.09	0.83 ± 0.004
CO ₂ uptake at LP	1.07 ± 0.05	2.07 ± 0.05	1.43 ± 0.06	0.48 ± 0.01
Band gap	0.75 ± 0.03	0.89 ± 0.07	0.94 ± 0.04	0.36 ± 0.03
H ₂ capacity	2.39 ± 0.08	10.30 ± 0.50	3.21 ± 0.07	0.95 ± 0.004
Xe uptake at HP	0.80 ± 0.01	1.21 ± 0.02	1.09 ± 0.01	0.90 ± 0.0005
Pore diameter	1.15 ± 0.03	3.61 ± 0.56	1.89 ± 0.15	0.79 ± 0.02
ASA	435.6 ± 27.5	327156 ± 35586	571.1 ± 31.5	0.72 ± 0.03
Density	0.16 ± 0.01	0.05 ± 0.003	0.23 ± 0.01	0.87 ± 0.01
CH ₄ DC	28.5 ± 0.50	1396.6 ± 25.5	37.4 ± 0.34	0.72 ± 0.001

“SI Table S14 Performance metrics for our model (accepting PXRD and precursors). The units for gas uptake, band gap, H₂ capacity, pore diameter, ASA, and density are mol/kg, eV, g/L, Å, m²/cm³, and g/cm³, respectively. This was run over 3 random seeds.”

Property	MAE	MSE	RMSE	SRCC
CH ₄ uptake at HP	1.35 ± 0.02	3.78 ± 0.16	1.94 ± 0.04	0.89 ± 0.01
CO ₂ uptake at LP	0.84 ± 0.02	1.41 ± 0.06	1.18 ± 0.03	0.69 ± 0.02
Band gap	0.376 ± 0.01	0.26 ± 0.03	0.51 ± 0.03	0.87 ± 0.01
H ₂ capacity	1.537 ± 0.06	4.31 ± 0.06	2.077 ± 0.01	0.98 ± 0.00
Xe uptake at HP	0.561 ± 0.002	0.582 ± 0.001	0.76 ± 0.00	0.95 ± 0.00
Pore diameter	1.08 ± 0.05	3.49 ± 0.16	1.86 ± 0.04	0.82 ± 0.01
ASA	360.2 ± 10.4	263866 ± 8837	513.61 ± 8.56	0.77 ± 0.004
Density	0.13 ± 0.004	0.04 ± 0.002	0.19 ± 0.005	0.92 ± 0.01
CH ₄ DC	24.4 ± 0.59	1062 ± 68.8	32.6 ± 0.01	0.79 ± 0.01

Reviewer Comment: We acknowledge the reading quality of the article and to original methodology and approach. We recommend the article to Nature Communications with minor revisions; see comments above.

Reviewer #2 Comments

Reviewer Comment: This paper presents a self-supervised multimodal model for predicting the properties and applications of metal-organic frameworks (MOFs) based on synthesis data, including powder X-ray diffraction (PXRD) patterns and chemical precursors. The authors demonstrate the model's effectiveness in generating property predictions and making application recommendations, including for future MOF discoveries. The manuscript provides a solid approach and promising results, but there are aspects that need further clarification and analysis to strengthen the findings. My concerns about the paper are as follows:

Reviewer Comment: 1. Effectiveness of Self-Supervised Learning: The loss curves for self-supervised training (flattening out at 40 epochs), combined with the small size of the pretraining dataset, make it unclear whether the approach is genuinely beneficial. While the authors report improved MAE for the pretrained models compared to training from scratch, the absence of standard deviations makes it hard to assess the reliability of these results. Were the test sets randomized? Was any k-fold cross-validation conducted? Is the improvement statistically significant? Pretraining typically demonstrates greater benefits on larger datasets, as shown in prior works, so it is not clear if the approach adds significant value here.

Authors Response: Thank you for highlighting this important point. We agree that including the standard deviation in the learning curve is crucial for demonstrating the impact of self-supervised learning (SSL). In the revised figure, we have incorporated k-fold cross-validation ($k = 5$) and recalculated the learning curves accordingly. To further quantify the reliability of our results, we have also evaluated the model across all MOF databases using three randomized seeds, ensuring that test sets remain consistent per seed (refer to the Reviewer #1 response for the regression results for these databases).

Our results show that self-supervised pretraining on CGCNN provides significant advantages, particularly for chemistry-dependent property predictions (e.g., Henry's coefficient of solubility for CO_2 and CO_2 uptake at low pressure). This benefit is particularly pronounced for smaller datasets such as ARABG-MOF ($N = 400$), where the pretrained model achieves a statistically significant improvement in Spearman's Rank Correlation Coefficient (SRCC) for CO_2 uptake (SRCC = 0.83) compared to training from scratch (SRCC = 0.64). The regression results supporting this are provided in the Supplementary Information. However, we acknowledge that for larger datasets (e.g., CoRE-MOF, BW20K, hMOF, and QMOF), the availability of sufficient training data makes training from scratch generally effective. We included the following short paragraph in the main text and updated the Figure S22 in the SI.

“In scenarios where we deal with small labelled datasets, a self-supervised pretraining workflow can improve the model performance, where the pretraining leverages large amounts of unlabelled data to learn generalizable representations that can be fine-tuned on the limited labelled samples. In our study, we pretrain our multimodal model taking PXRDs and precursors against crystal structures available from large MOF databases. We show in the SI that this approach improves the performance of the model on small datasets, such as ARABG, and make the method applicable for the cases when large amount of data is not available.”

Figure S22. Learning curves showcasing impact of self-supervised learning (SSL) on a small dataset A comparison was made between the pre-trained and scratch models of our model (panels (a) and (b) showcasing mean absolute error (MAE) and spearman rank correlation coefficient (SRCC) respectively), showcasing its influence on small datasets - particularly in very small data regimes. Furthermore, comparisons were made between benchmarked models (MOFormer, CGCNN, descriptors - panels (c) and (d)) and our pre-trained model, showcasing its advantage at low data regimes on small datasets such as ARABG-DB.

“Table S5: Comparison of scratch and pretrained models on CoRE-MOF across 3 random seeds.”

Property	MAE	MSE	RMSE	SRCC
Scratch Model				
CH ₄ uptake @ HP	1.39 ± 0.04	4.25 ± 0.53	2.06 ± 0.12	0.89 ± 0.01
CO ₂ uptake @ LP	0.87 ± 0.03	1.46 ± 0.12	1.21 ± 0.05	0.67 ± 0.02
Pore diameter	1.08 ± 0.05	3.49 ± 0.16	1.86 ± 0.04	0.82 ± 0.01
Density	0.13 ± 0.004	0.04 ± 0.002	0.19 ± 0.005	0.92 ± 0.01
LogKH_CO ₂	0.61 ± 0.03	0.66 ± 0.02	0.81 ± 0.01	0.70 ± 0.01
CH ₄ DC	25.2 ± 1.23	1112 ± 96.0	33.3 ± 1.44	0.77 ± 0.01
Pretrained Model				
CH ₄ uptake @ HP	1.35 ± 0.02	3.78 ± 0.16	1.94 ± 0.04	0.90 ± 0.01
CO ₂ uptake @ LP	0.84 ± 0.02	1.41 ± 0.06	1.18 ± 0.03	0.70 ± 0.02
Pore diameter	1.15 ± 0.11	4.02 ± 0.90	1.99 ± 0.22	0.82 ± 0.01
Density	0.13 ± 0.004	0.036 ± 0.002	0.19 ± 0.006	0.91 ± 0.01
LogKH_CO ₂	0.57 ± 0.02	0.62 ± 0.03	0.78 ± 0.02	0.72 ± 0.02
CH ₄ DC	24.4 ± 0.59	1062 ± 68.8	32.6 ± 0.01	0.79 ± 0.01

“Table S6: Comparison of scratch and pretrained models on BW20K run over 3 random seeds.”

Property	MAE	MSE	RMSE	SRCC
Scratch Model				
CH ₄ uptake @ HP	0.97 ± 0.04	1.67 ± 0.15	1.29 ± 0.06	0.97 ± 0.00
CO ₂ uptake @ LP	0.39 ± 0.05	0.33 ± 0.09	0.57 ± 0.07	0.84 ± 0.04
Pore diameter	0.86 ± 0.05	1.53 ± 0.19	1.23 ± 0.07	0.95 ± 0.00
Density	0.07 ± 0.01	0.01 ± 0.00	0.09 ± 0.02	0.97 ± 0.01
LogKH_CO ₂	0.25 ± 0.01	0.13 ± 0.01	0.35 ± 0.01	0.88 ± 0.00
Pretrained Model				
CH ₄ uptake @ HP	0.76 ± 0.01	1.05 ± 0.03	1.02 ± 0.01	0.98 ± 0.00
CO ₂ uptake @ LP	0.31 ± 0.00	0.21 ± 0.00	0.46 ± 0.00	0.90 ± 0.00
Pore diameter	0.72 ± 0.01	1.08 ± 0.04	1.04 ± 0.02	0.97 ± 0.00
Density	0.05 ± 0.00	0.01 ± 0.00	0.07 ± 0.00	0.99 ± 0.00
LogKH_CO ₂	0.23 ± 0.00	0.10 ± 0.01	0.33 ± 0.01	0.90 ± 0.01

“Table S7: Comparison of scratch and pretrained model on ARABG-DB (N = 400 entries) run over 3 random seeds.”

Property	MAE	MSE	RMSE	SRCC
Scratch Model				
CH ₄ uptake @ HP	1.24 ± 0.15	3.64 ± 1.67	1.86 ± 0.42	0.92 ± 0.03
CO ₂ uptake @ LP	0.32 ± 0.06	0.28 ± 0.09	0.52 ± 0.08	0.64 ± 0.05
Pore diameter	1.91 ± 0.20	6.54 ± 0.80	2.55 ± 0.16	0.90 ± 0.03
Density	0.14 ± 0.04	0.04 ± 0.02	0.18 ± 0.05	0.87 ± 0.09
LogKH_CO ₂	0.26 ± 0.02	0.11 ± 0.02	0.33 ± 0.03	0.76 ± 0.07
Pretrained Model				
CH ₄ uptake @ HP	1.11 ± 0.13	2.31 ± 0.56	1.51 ± 0.18	0.95 ± 0.01
CO ₂ uptake @ LP	0.22 ± 0.03	0.14 ± 0.06	0.37 ± 0.07	0.83 ± 0.01
Pore diameter	1.20 ± 0.12	3.24 ± 1.62	1.75 ± 0.43	0.95 ± 0.01
Density	0.11 ± 0.01	0.02 ± 0.001	0.13 ± 0.004	0.94 ± 0.001
LogKH_CO ₂	0.19 ± 0.01	0.07 ± 0.01	0.26 ± 0.03	0.84 ± 0.03

“Table S.8: Comparison of scratch and pretrained models on QMOF (band gap) and hMOF (hydrogen capacity and Xe uptake at HP). For QMOF, it was run over 3 random seeds, whereas hMOF was run over 2 random seeds (due to computational restrictions).”

Property	MAE	MSE	RMSE	SRCC
Scratch Model				
Band gap	0.39 ± 0.02	0.28 ± 0.03	0.53 ± 0.03	0.85 ± 0.01
H ₂ Capacity	1.88 ± 0.07	6.13 ± 0.58	2.47 ± 0.12	0.96 ± 0.00
Xe uptake @ HP	0.58 ± 0.03	0.61 ± 0.03	0.78 ± 0.02	0.95 ± 0.00
Pretrained Model				
Band gap	0.38 ± 0.01	0.26 ± 0.03	0.51 ± 0.03	0.87 ± 0.01
H ₂ Capacity	1.54 ± 0.06	4.31 ± 0.06	2.08 ± 0.01	0.98 ± 0.00
Xe uptake @ HP	0.56 ± 0.00	0.58 ± 0.00	0.76 ± 0.00	0.95 ± 0.00

Reviewer Comment: Figure 2 is slightly difficult to interpret as presented. Including a table to accompany the figure would make the results easier to understand and improve clarity.

Authors Response: The authors agree with this suggestion. Please refer to Figure 2 under Reviewer #1 comment for an updated figure.

Reviewer Comment: Descriptor-based models perform better on most tasks in Figure 2. Have these descriptors been used in MOF recommendation systems? Would they give more accurate results given their good performance on tasks in Figure 2?

Authors Response: To the best of our knowledge, descriptors (revised autocorrelations - RACs and geometric descriptors) have not been used in MOF recommendation systems. Although models based on these descriptors may offer enhanced performance, they are not aligned with the primary goal of our study, which is to bridge synthesis with application. Obtaining such descriptors requires a fully resolved and computation-ready crystal structure. In practice, such data is not readily available upon MOF synthesis. One would need to resolve the MOF's crystal structure, process it into a computation-ready format—by removing solvents, resolving partial occupancies, and adding hydrogen—and then compute the descriptors using tools like molSimplify and Zeo++. While the descriptor calculations themselves are straightforward, preparing the crystal structures is both challenging and time-consuming. Our approach, in contrast, offers a shortcut, while maintaining the performance and accuracy, by enabling experimentalists to simply upload the readily available precursors and PXRD data immediately after synthesis, thereby receiving timely recommendations. This accelerates the discovery of MOF applications by leveraging data that is available at the point of synthesis. We included the following text in the revised version to better clarify our goal in this study:

“We note that while descriptor-based models offer slightly better performance in property prediction, they require preprocessed crystal structures in a computation-ready format (e.g., solvent removal and handling of partial occupancy), which can be challenging. In contrast, our approach leverages PXRD and precursor information, which are readily available upon synthesis, allowing us to provide accurate recommendations for both chemistry-reliant and geometry-reliant properties and applications without the need for extensive structural preprocessing.”

Reviewer Comment: 4. The “time-travel” experiment is promising but raises questions about the similarity between the training and test sets. How distinct are the MOFs in the test set compared to those in the training set? Are the results reflective of the model's true generalizability, or is it leveraging patterns from previously seen data? An analysis of embedding similarity between test points and their closest training samples could help clarify this.

Authors Response: Thank you for raising this point. We conducted a similarity analysis between the embeddings of the train and test sets, and we have included the results in the Supplementary Information. Our findings indicate that the model does not merely memorize patterns from the training set and apply them to the test set, as evidenced by the low cosine similarity between their embeddings.

We would also like to emphasize that the purpose of the time-travel experiment was to demonstrate that if the model is trained on all MOFs available today, it can be used to predict potential applications for newly synthesized MOFs in the future. This highlights the model's utility in accelerating the discovery of novel material applications beyond their originally intended use. As in this case we are in the deployment phase of the model, it is not necessarily harmful to have similarity as we do not report performance metrics.

Figure S.16. Similarity analysis between model embeddings for train and test sets for “time-travel” model For a baseline, a simple random train/test split was done and used to train a model on CO₂ uptake at low pressure, with the train and test model embedding cosine similarities computed in panel (a). For panel (b), a similar computation was done on the train (before 2017) and test (after and including 2017) sets for the “time-travel” model. Both splits are shown to have low mean cosine similarity (around 0.35 for both).

Reviewer Comment: 5. The justification for specific thresholds in the recommendation system (e.g., methane uptake values) is missing. Providing reasoning for these thresholds in the SI would improve the clarity of the recommendations.

Authors Response: The authors would like to point out that the thresholds in the recommendation system, alongside the sources in which those thresholds were obtained, are in the Supplementary Information. However, we also acknowledge that stating the reasoning for these thresholds would improve the clarity of that section, and we have introduced an enumerated description below that table justifying the threshold values.

“The following describes the reasoning for the threshold selections:

1. *Methane storage: Anything above 12 mol/kg is considered very promising, as that is approximately the uptake at 65 bar for HKUST-1, which is a commonly used MOF in gas storage applications. 7.93 mol/kg was chosen, as that was the cutoff to retrieve the top 30 percent of MOFs (in terms of methane uptake at 65 bar) available in the used CoRE-MOF database;*
2. *Hydrogen storage: For balanced hydrogen capacity, IRMOF-20 is known to be a “record-holder”, with an available capacity of 33.4 g H₂/L - hence why this criteria was used for flagging very promising MOFs for hydrogen capture. 27 g/L was chosen, as it was the cutoff to retrieve the top 30 percent of MOFs (in terms of hydrogen capacity) available in the used hMOF database;*
3. *Xenon storage: Due to the lack of experimental data online for xenon storage in MOFs, the top 10 and 30 percent of xenon uptakes in the hMOF database were chosen to be the thresholds to flag a MOF as “very promising” and “promising” respectively;*

4. *Carbon capture: Mahajan et al. (2022) stated that for a promising candidate for carbon capture, anything greater than 2 mmol CO₂/g adsorbent is acceptable. 3.6 mol/kg was calculated from retrieving the top 10 percent of MOFs from the available CoRE-MOF database;*
5. *Direct-air capture: Due to the lack of available experimental data online, the top 10 and 30 percent of the Henry coefficient of solubility for CO₂ were taken to be the thresholds for a very promising and promising MOF respectively;*
6. *Band gap: Zhang et al. (2024) stated that MOFs with band gap between 1-3 eV exhibit semiconductor behaviour and are appropriate to use as sensors, microelectronics, etc.. Anything below this range i.e. 1 eV is chosen to be a very promising MOF, band gap wise.*

”

Reviewer Comment: 6. While robustness testing is included, the impact of significant noise or distortions in PXRD data is not fully explored. For example, the authors state, ”In this inconsistent case, the difference between the simulated and experimental PXRDs is pronounced; the experimental PXRD exhibits noticeable noise and multiple significant peaks compared to the simulated PXRD.” How is ”pronounced” defined in this context? Quantifying these differences or providing clear metrics would improve clarity and allow for a better assessment of the model’s performance under noisy or distorted conditions.

Authors Response: Thank you for this insightful comment. To address this concern, we have conducted a robustness analysis of the model under two noise scenarios: 1) Low crystallinity materials – In cases where the material exhibits broad peaks due to low crystallinity, we introduced Gaussian noise to the PXRD data to simulate these distortions. 2) Impure phases – To mimic situations where the synthesized material is not phase pure, we artificially introduced an additional MOF’s PXRD pattern into the original material’s PXRD.

To quantify the model’s robustness under these conditions, we report the F1-score across varying noise levels in the Supplementary Information. These results provide a clearer assessment of how noise and distortions impact model performance.

“An analysis was done to assess the robustness of the model on different forms of noise and to see at what point the model robustness fails. Two scenarios were considered:

1. *Low crystallinity materials – In cases where the material exhibits broad peaks due to low crystallinity, we introduced Gaussian noise to the PXRD data to simulate these distortions. The modified PXRD, denoted as C'_{PXRD} , is computed as:*

$$C'_{PXRD} = C_{PXRD} + \epsilon |N(\mu = 1, \sigma^2 = 1)|$$

where $N(\mu, \sigma^2)$ represents a Gaussian distribution with mean μ and variance σ^2 , and ϵ is a ”noise factor” ranging from 0 to 1.

2. *Impure phases – To mimic situations where the synthesized material is not phase pure, we artificially introduced an additional MOF’s PXRD pattern into the original material’s PXRD. The modified PXRD (C'_{PXRD}) can be expressed as:*

$$C'_{PXRD} = C_{PXRD} + \epsilon C_{ref}$$

Figure S.23 F1-scores for the impact of noise in PXRD data on model robustness The two case studies looked at are when Gaussian noise is added (red) and when a reference PXRD (CSD reference code: NUHQOY) was added to the PXRD data (red).

where C_{ref} and ϵ are the reference PXRD and a "noise factor" ranging from 0 to 1.

Figure S.21 shows the F1-scores for the model recommendation classifications (i.e. how accurately the model can correctly predict the application for methane storage), while increasingly adding more noise. For the first case study, the model is shown to be quite robust until around 5-10 percent of Gaussian noise is added, in which the model robustness rapidly decreases and flattens out at F1-scores close to 0. For the second case study, however, we can see that the model robustness is quite acceptable even at 20 percent noise added. However, from there onwards, the performance gradually decreases - although not at a similar rate as the first case study."

Reviewer Comment: There is a lack of comparison to similar baselines on property prediction benchmarks, such as MOFormer or CGCNN.

Authors Response: Thank you for this comment. We have included benchmarking for various geometric, chemistry-reliant and quantum chemical property predictions on MOFormer and CGCNN (please see Figure 2 in main text - left panel, and Figure 2 in this document). Furthermore, an ablation study showcasing the need for multimodality is included in the right panel of the figure. The main text was modified to highlight these changes better as following:

"Specifically, we benchmark our model against three state-of-the-art approaches: 1) Descriptor-based ML, which uses geometric and chemical descriptors (e.g., surface area, pore volume, and revised autocorrelation (RACs); see SI for details), 2) Crystal Graph Convolutional Neural Network (CGCNN), which represents materials using a crystal graph, and 3) MOFormer, a model that takes a text-based representation of MOFs (MOFid) as input. While MOFormer uses a string representation, this encoding requires crystal structure information (e.g., MOF topology and metal cluster

details), making it dependent on structural processing. Our model performs comparably to crystal structure-based models across all property types. Specifically for geometric properties, such as surface area, our model outperforms both CGCNN and MOFormer. The descriptor-based ML model achieves perfect scores for surface area and density, as these are directly provided as inputs. For geometry-reliant properties, such as high-pressure adsorption of CH₄ and Xe, our model achieves accuracy comparable to descriptor-based approaches. Finally, for chemistry-reliant and quantum-chemical properties, our model performs on par with crystal structure-based models (Figure 2).”

“It is insightful to compare the performance of our multimodal model with a model that relies solely on the chemical precursor and another model that relies solely on the PXRD. The results of the ablation study are shown on the right panel of Figure 2. The results show that a model only accepting chemical precursors is inadequate for capturing the global structure of the MOF, scoring particularly low in geometric-reliant and pure geometric properties. On the other hand, a model only accepting PXRDs capture the global structure of the MOF well, but fails to capture the local environment, scoring low in chemistry-reliant and quantum chemical properties such as CO₂ uptake at LP and band gap. This confirms that the PXRD is critical and provides sufficient information about the geometry of a MOF, whereas the precursor aids in capturing information about the local environment. Achieving high accuracy in all three categories using only the information available at the time of MOF synthesis underscores the importance of multimodality in property prediction tasks. No single modality can fully encapsulate the complexities of chemistry and geometry in MOFs; thus, integrating both PXRD and precursor strings facilitates accurate prediction of material properties. The accuracy of the multimodal model establishes the foundation for using its predictions to recommend MOFs for various applications.”

Reviewer #3 Comments

The code is well-written and accessible to both researchers with strong and limited programming experience. However, to further improve accessibility for a broader audience and enhance reproducibility, the authors should:

Reviewer Comment: - Provide a script to directly load the model and generate predictions.

Authors Response: Revised accordingly in updated repository (see predictions.ipynb notebook for an example).

Reviewer Comment: Add a tutorial on converting experimental PXRD data into a format compatible with the model.

Authors Response: Revised accordingly - please see experimental.ipynb notebook for an example.

Reviewer Comment: Modify the tutorial to include the recommended Python version, as it currently fails to work if the Python version is not specified.

Authors Response: Revised accordingly - please see the README of the repository for an installation guide, which now includes the recommended Python version.

Reviewer Comment: Clarify how uploaded data on the web app is handled—whether it is stored in an external database or removed after a certain period—this is especially critical for newly synthesized MOFs or data from companies.

Authors Response: The authors would like to point out that there is no external database that stores submitted MOF PXRD patterns/precursor strings or any data inputted into our web application. However, we acknowledge this concern, and will add a comment in our README for the web application repository that no form of data is stored or taken by us.

Response to Reviewers for “Connecting metal-organic framework synthesis to applications with multimodal machine learning”

We thank the reviewers for their valuable comments and constructive feedback, which have helped us improve the quality of our manuscript; we have addressed each point in detail below. The reviewers’ comments are in gray, the authors’ response in blue and changes in the manuscript in green.

Reviewer #2 Comments

Reviewer Comment: 1. In the main section authors mention: ”This pretraining enables the model to achieve high accuracy across a range of chemical, geometric, and quantum-chemical property predictions.” However, in their response letter they have stated: ”However, we acknowledge that for larger datasets (e.g., CoRE-MOF, BW20K, hMOF, and QMOF), the availability of sufficient training data makes training from scratch generally effective.” This limitation should be explicitly acknowledged in the main text, as self-supervised learning shows no tangible benefit for larger datasets. I would suggest that the authors also acknowledge this limitation in the main section or any other section of the manuscript.

Authors Response: The authors agree with the suggestion made by the reviewer and have adjusted the main text such that we acknowledge that self-supervised learning leads to significant model performance improvement in low data regimes only.

“This pretraining enables the model to achieve high accuracy across a range of chemical, geometric, and quantum-chemical property predictions on low data regimes.”

“In our study, we pretrain our multimodal model taking PXRDs and precursors against crystal structures available from large MOF databases. We show in the SI that this approach improves the performance of the model on small datasets, such as ARABG, and make the method applicable for the cases when large amount of data is not available. However, it should be noted that the model performance has very negligible improvement when the pretrained model is fine-tuned on large MOF databases such as CoRE-MOF, BW20K, QMOF and hMOF.”

Reviewer Comment: 2. The authors have addressed all my other concerns